# Plastic Changes in Pain and Motor Network Induced by Chronic Burn Pain

**DOI:** 10.3390/jcm10122592

**Published:** 2021-06-11

**Authors:** So Young Joo, Chang-hyun Park, Yoon Soo Cho, Cheong Hoon Seo, Suk Hoon Ohn

**Affiliations:** 1Department of Rehabilitation Medicine, Hangang Sacred Heart Hospital, College of Medicine, Hallym University, Seoul 07024, Korea; anyany98@naver.com (S.Y.J.); hamays@hanmail.net (Y.S.C.); chseomd@gmail.com (C.H.S.); 2Center for Neuroprosthetics and Brain Mind Institute, Swiss Federal Institute of Technology (EPFL), 1202 Geneva, Switzerland; park.changhyun@hanmail.net; 3Department of Physical Medicine and Rehabilitation, Hallym University Sacred Heart Hospital, Hallym University College of Medicine, Anyang 14068, Korea

**Keywords:** chronic pain, burn, cerebral blood volume

## Abstract

Musculoskeletal diseases with chronic pain are difficult to control because of their association with both central as well as the peripheral nervous system. In burn patients, chronic pain is one of the major complications that cause persistent discomfort. The peripheral mechanisms of chronic pain by burn have been greatly revealed through studies, but the central mechanisms have not been identified. Our study aimed to characterize the cerebral plastic changes secondary to electrical burn (EB) and non-electrical burn (NEB) by measuring cerebral blood volume (CBV). Sixty patients, twenty with electrical burn (EB) and forty with non-electrical burn (NEB), having chronic pain after burn, along with twenty healthy controls, participated in the study. Voxel-wise comparisons of relative CBV maps were made among EB, NEB, and control groups over the entire brain volume. The CBV was measured as an increase and decrease in the pain and motor network including postcentral gyrus, frontal lobe, temporal lobe, and insula in the hemisphere associated with burned limbs in the whole burn group. In the EB group, CBV was decreased in the frontal and temporal lobes in the hemisphere associated with the burned side. In the NEB group, the CBV was measured as an increase or decrease in the pain and motor network in the postcentral gyrus, precentral gyrus, and frontal lobe of the hemisphere associated with the burn-affected side. Among EB and NEB groups, the CBV changes were not different. Our findings provide evidence of plastic changes in pain and motor network in patients with chronic pain by burn.

## 1. Background

Pain is a perceptual phenomenon caused due to the integration and modulation of several neuronal, psychological, and cognitive processes and it comprises sensory-discriminative, emotional, and cognitive dimensions [1]. Chronic pain from musculoskeletal disease involves changes in the peripheral as well as the central nervous system [2,3,4] and is consequently associated with unique functional and structural brain changes [5,6,7,8]. Chronic pain after burn is caused by multiple changes in the skin, muscle, and nerve by direct thermal injury, vascular occlusion, and post-injury edema [9,10]. Hypertrophic scar change of burn wound induces increased sensitivity of nociceptors, which are the smallest unmyelinated and lightly myelinated primary afferent nerve fibers in the peripheral nerves [10,11,12]. Like other chronic pain, burn-induced pain is thought to be accompanied by changes in the central nervous system as well as the peripheral nervous system. However, research on burn-induced pain and changes in the central nervous system is still in its nascent stage. Considering the effect of an antiepileptic drug on burn-induced pain, it is presumed that chronic pain after burn is related to the central nervous system. The authors hypothesize that the comparative study of cerebral blood volume (CBV) changes induced by electrical burn (EB) and non-electrical burn (NEB) could give a clue in understanding the pain mechanism related to the brain. This is because of the possibility of a different mechanism of pain between the EB and the NEB, in that the high-voltage-electrical current can cause pain by damage to the nervous system, but this is unlikely in the NEB [5,8,13].

The CBV, which is observed using magnetic resonance imaging (MRI), is considered to be a hemodynamic variable that is highly correlated with oxygen metabolism, representing the fraction of cerebral tissue volume occupied by blood at a given time point [14,15,16]. The purpose of this study was to analyze the changes in CBV of the cerebral pain network in patients with chronic pain caused by electrical burns (EB) and non-electrical burns (NEB).

## 2. Methods

The study protocol was approved by the local Institutional Review Board (No. HG2019-025), and was registered by the Clinical Trials Registration (NCT04125576). Written informed consent was obtained from all participants.

### 2.1. Clinical Subjects

This is a single-center, prospective observational study. Inclusion and exclusion criteria were predefined to include the study participants. Inclusion criteria were (i) age: ≥18 years, (ii) patients with a certain intensity of pain in the burn area: at least a five rating on the 10-point visual analogue scale (VAS) despite drug treatment and physical therapy, among the patients admitted to the Department of Rehabilitation Medicine because of sustained burn pain after skin epithelialization due to acute therapy, (iii) partial or full-thickness burns that healed spontaneously or required skin grafting, and (iv) burn on either right or left side of the body. Exclusion criteria were (i) history of brain disease or injury, (ii) a history of cardiac arrest, (iii) pain resulting from other possible causes (e.g., osteoarthritis, neuromuscular disease, or peripheral neuropathy) as confirmed via medical diagnostic tools (radiography, ultrasonography, computed sonography, or MRI), (iv) other forms of persistent pain lasting more than 3 months, (v) amputation, (vi) a score of 8 or higher on the Korean version of the Hamilton depression rating scale (HDRS) [17], (vii) psychiatric disorders, (viii) diabetes mellitus, (ix) abnormal renal function, (x) contraindications for MRI, (xi) or pregnancy. Discontinuation of pain medication was not possible for ethical reasons. To exclude drug effects as much as possible, the dose of drugs already being administered for pain control was maintained. Antiepileptic and antidepressant drugs such as gabapentin or amitriptyline, which were used for pain control and may affect MRI imaging [18,19], were maintained at the same dose until one week before the start of the experiment. The patients with extended-release morphine therapy were excluded [20]. Additionally, considering the importance of an intact blood-brain barrier for steady-state gadolinium-enhanced MRI, one radiologist carefully reviewed the pre-enhancement T1-weighted, post-enhancement T1-weighted, T2-weighted, and diffusion-weighted MR images of all participants. Patients with suspected parenchymal injury were excluded from the study. Applying the same exclusion criteria, we recruited 20 healthy, sex and age-matched control participants. Before they participated in the study, none of the control participants had experienced persistent pain for more than 1 week.

### 2.2. Clinical Assessments

Patients were asked to score their mean pain intensity by VAS and Brief Pain Intensity (BPI). A score of 0/10 indicated stimulus perception but no pain. A 10/10 score was defined as intolerable pain. The BPI consists of the sensory dimension, which indicates the intensity of pain, and the reactive dimension, which measures the decrease in function due to pain [21]. Depressive mood was assessed using the previously validated Korean version of the HDRS [17]. The HDRS was administered to all participants by a single licensed psychologist.

### 2.3. MRI Acquisition and CBV Mapping

All MRI images were obtained using a 3.0 T magnetic resonance scanner (MAGNETOM Skyra^TM^, Siemens, Erlangen, Germany) using an established steady-state gadolinium-enhanced MRI technique [22]. Two high-resolution T1-weighted images (repetition time/echo time = 2000.0 ms/2.26 ms, flip angle = 15°, field of view = 256 mm × 256 mm, in-plane resolution = 1 × 1 mm, slice thickness = 1 mm) were acquired for each participant, once before administration of a standard intravenous dose of gadolinium contrast agent (0.1 mmol/kg Dotarem^®^; Guerbet, Villepinte, France) and another four minutes after gadolinium administration. T2-weighted and diffusion-weighted MR images were taken to exclude patients with suspected parenchymal injury. T2-weighted MRI was obtained according to the following parameters: TR/TE = 8000/105 ms, field of view = 220 (AP) 144 (FH) 214 (RL) mm. Thirty-one images, all except one with high diffusion weighting at *b* value = 1000 sec/mm^2^ and one without diffusion weighting, were acquired with a single-shot diffusion-weighted echo-planar imaging sequence. Each image included 65 to 70 axial slices with the following parameters: TR/TE = 9400.0 ms/0.95 ms, flip angle = 90°, field of view = 220 mm × 220 mm, in-plane resolution = 1.72 mm × 1.72 mm, slice thickness = 2.30 mm.

Processing of CBV mapping data was performed using SPM12 software (The Wellcome Centre for Human Neuroimaging, UCL Queen Square Institute of Neurology, London, UK), as in previous studies [22,23]. Two structural images from each participant were processed following routines specified in SPM12 software. The post-enhanced image was coregistered to the pre-enhanced image, and both images were normalized to the same coordinate frame as the Montreal Neurological Institute template brain. Briefly, pre-contrast and post-contrast images were spatially transformed to the same standard space, and then a map of contrast-induced signal difference ratios was acquired as (post-contrast signal—pre-contrast signal)/(maximum signal difference in the superior sagittal sinus) × 100. For patients with injuries on the left extremities, the map was flipped around the mid-sagittal axis [22].

#### Statistical Analysis

To examine pretreatment homogeneity among participating groups, a Kruskal–Wallis test was used for age after normality test. To examine pretreatment homogeneity between EB and NEB groups, a Mann–Whitney *U* test was used for total body surface area (TBSA), duration between injuries and MRI acquisition, VAS score, BPI score, and HRDS score after normality test. Fisher’s exact test was used to assess sex differences between the EB and NEB groups.

The main study outcomes were group differences in CBV, and the association between CBV changes and the pain intensity. These analyses were performed by applying a general linear model to each voxel with the same coordinates [24]. For each map, voxel-wise comparisons between groups were evaluated by adjusting for the effects of individual participants’ age, sex, and degree of depression. Group differences with a *p* value of 0.05 corrected for multiple comparisons with threshold-free cluster enhancement were considered statistically significant [25]. Evaluated data were analyzed with SPSS version 23.0 (IBM, Armonk, NY, USA).

## 3. Results

### 3.1. Clinical Features

Twenty EB patients and forty NEB patients were enrolled based on the inclusion/exclusion criteria. Age and sex were not different among groups (Table 1). TBSA, duration between burn and MRI acquisition, VAS, BPI (sensory dimension and reactive dimension), and HDRS scores were not different between the EB group and the NEB group.

#### CBV Mapping of Chronic Pain after Burns

In all burn patients, there was a CBV increase in the postcentral gyrus of the hemisphere associated with the burned limb (*p* = 0.045), and there was CBV decrease in the inferior frontal gyrus (*p* = 0.004), middle frontal gyrus (*p* = 0.005), precentral gyrus (*p* = 0.008), superior temporal gyrus (*p* = 0.031), anterior lateral temporal lobe (*p* = 0.026), inferior middle temporal gyrus (*p* = 0.032), posterior temporal lobe (*p* = 0.031), and insula (*p* = 0.035) of the hemisphere associated with the burned limb compared to healthy controls (Figure 1A and Table 2). The EB group exhibited a decrease in CBV in the anterior lateral temporal lobe (*p* = 0.014), inferior middle temporal gyrus (*p* = 0.021), inferior frontal gyrus (*p* = 0.044), middle frontal gyrus (*p* = 0.045), and superior temporal gyrus (*p* = 0.038) of the hemisphere associated with the burned limb in comparison to healthy controls (Figure 1B and Table 3). The NEB group exhibited an increase in CBV in the postcentral gyrus (*p* = 0.017) and superior parietal gyrus (*p* = 0.018) of the hemisphere associated with the burned limb and a decrease in CBV in the inferior frontal gyrus (*p* = 0.017), middle frontal gyrus (*p* = 0.023), precentral gyrus (*p* = 0.031), and middle frontal gyrus (*p* = 0.046) of the hemisphere associated with the burned limb (Figure 1C and Table 4). There was no statistically significant difference in CBV between the EB group and the NEB group.

## 4. Discussion

The peripheral mechanism of chronic pain associated with burn injury is well known. However, there are few studies on the central mechanisms of chronic pain caused due to burns. The purpose of this study was to identify the changes in pain network related to burn-induced chronic pain by analyzing the CBV of patients with EB and NEB due to the possibility of a different mechanism of pain between these two types of burns [25,26].

The pain network comprises sensory-discriminative, emotional, and cognitive dimensions. The sensory-discriminative dimension perceives the sense of intensity, location, quality, and duration of pain and includes the lateral thalamic system, primary and secondary somatosensory cortex, and insula cortex [6,27]. When the afferent sensory nerve function is altered, the activity of the sensory-discriminative dimension is deactivated [28] or activated [29,30] to compensate for the altered afferent signals and suppress the pain. The emotional dimension is associated with negative emotion or memory of pain and includes the medial thalamic system, orbitofrontal cortex, anterior cingulated cortex, amygdala, and the insula cortex. The different dimensions of pain intensity are associated with different areas of the brain; sensory-discriminative dimension, emotional dimension, and motor dimension are related to the premotor and supplementary motor area, modulatory dimension is related to the frontal and parietal lobes, and autonomic nervous system dimension is related to the parietal lobe, insular cortex, and anterior cingulate cortex [31]. Parietal lobe is considered to be the region of brain dedicated to pain perception and modulation [31,32,33,34,35].

In this study, CBV mapping in the burn patients showed changes in the sensory-discriminative dimension of pain network indicated by an increased CBV in the postcentral gyrus and a decreased CBV in the temporal lobe and insular cortex compared to the controls. Moreover, the CBV decreased in the frontal lobe and precentral gyrus, indicating decreased motor network involvement. This means that chronic burn-induced pain involves plasticity in the pain network as well as the motor network. The changes in the motor network were expected to be due to the limited motion of the burn affected side. Compared to the controls, sensory-discriminative dimension of pain network was decreased in the EB group indicated by decreased CBV in the temporal lobe and the motor network in the frontal lobe. In the NEB group, the CBV increased in the postcentral gyrus and the parietal lobe, indicating increase in the sensory-discriminative dimension of pain network and a decrease in motor network as seen in the frontal lobe. We found that both the EB group and the NEB group had similar compensatory changes in the sensory-discriminative dimension and motor network.

We examined the difference in the CBV map of the EB group and the NEB group. Because all patients with EB were injured by high-voltage-electrical current (10,000~20,000 volts) and the brain is likely to be affected by high-voltage-electrical current directly through the peripheral nerve and spinal cord [13,36,37,38,39], this might have an influence on the pain network. However, we found that the pain network alterations were not different in the EB group and the NEB group in this study. Moreover, the pattern of CBV change was somewhat similar to each other in the pain discriminative network and the motor network. The cerebral pain network of EB group in this study is less likely to be directly affected by the electrical current through the peripheral nerve and spinal cord, which is similar to the results of other studies [22,23]. If the electrical current had an effect on the brain, watershed zones vulnerable to insufficient blood flow must have been damaged first. However, these areas were spared in this study and other studies, suggesting that the network changes did not occur because of electrical injury but were rather related to neuroplasticity associated with chronic burn pain.

We observed changes in the CBV of both pain discriminative network and motor network and the altered CBV regions were in the sensory-discriminative dimension rather than the emotional dimension or the cognitive dimension, though changes in the emotional neural network are usually seen in chronic pain. Patients with phantom limb pain after leg amputation due to high-voltage-electrical injury were suffering from both pain and depression, and therefore, changes in CBV were seen in the emotional dimension [22]. Patients with amputation due to EB would have felt more shock and anxiety; moreover, patients with HDRS 8 or over were not included in this study. According to the inclusion and exclusion criteria mentioned in the method section, for the purpose of observing the effect of only burn scars without severe dysfunction or depression, we could not observe the difference between electrical burn and non-electrical burn groups. This can be the cause of change in the sensory-discriminative dimension of pain network in chronic burn-induced pain.

The present study has a few limitations. First, we only included 60 patients and this small sample size may have weakened the statistical power. Second, patients reported uncontrolled pain and therefore could not discontinue medication before enrollment in this study. We cannot therefore exclude the possibility that pain medication might have affected brain imaging results. Third, patients who underwent amputation due to severe damage caused by burns were excluded. Therefore, a further study on the cerebral pain network analysis according to the degree of burn is necessary to confirm the exact mechanisms of chronic pain due to burns.

Our findings provided clear evidence for plastic changes in pain and motor network in patients with chronic pain by burn. In the patients who had severe pain and no depression, the plastic changes in the pain network were limited to the sensory-discriminative dimension and this factor can be considered for the management of similar cases of pain, that is a focal brain stimulation targeting the sensory-discriminative and motor dimension.

## Figures and Tables

**Figure 1 jcm-10-02592-f001:**
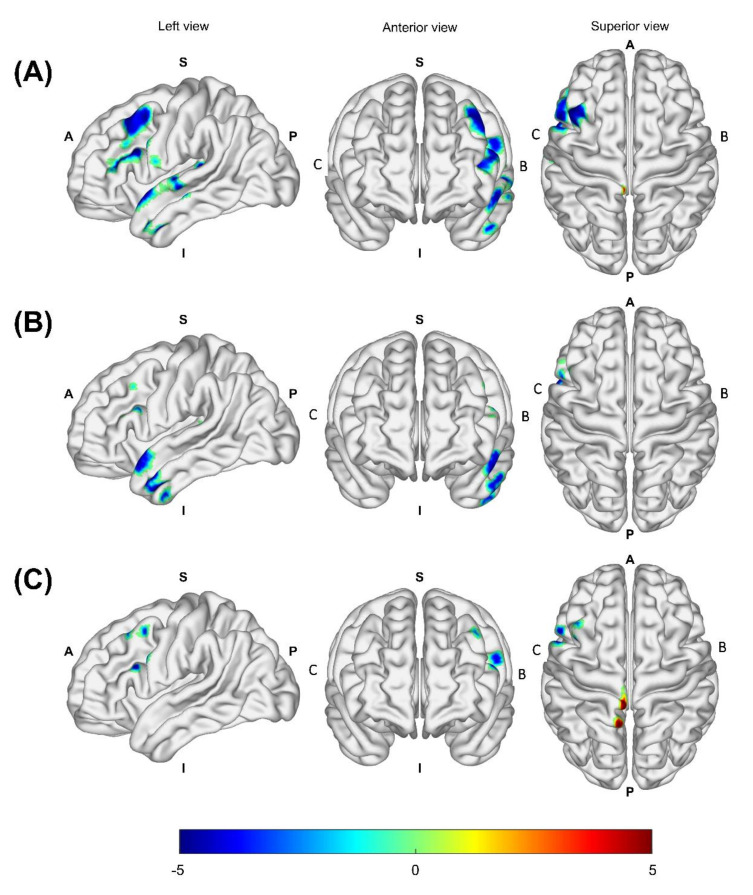
Mapping of patients’ versus controls’ CBV status as adjusted for sex and age. The CBV was measured as an increase and decrease in the pain and motor network including postcentral gyrus, frontal lobe, temporal lobe, and insula in the hemisphere associated with burned limbs in all burn patients (**A**). In the EB group, the CBV was decreased in the pain and motor network including the frontal and temporal lobe in the hemisphere associated with burned limbs (**B**). In the NEB group, the CBV was measured as an increase and decrease in the pain and motor network including the postcentral gyrus, precentral gyrus, and frontal lobe in the hemisphere associated with burned limbs (**C**). Areas with increased CBV are red-yellow and areas with decreased CBV are blue-light blue. Abbreviations: CBV, cerebral blood volume; EB, electrical burn; NEB, non-electrical burn; B, hemisphere associated with burned limb; C, hemisphere associated with contralateral intact limb; A, anterior; P, posterior; S, superior; I, inferior.

**Table 1 jcm-10-02592-t001:** Demographic data of subjects.

Variable	EB (*n* = 20)	NEB (*n* = 40)	Control (*n* = 20)
Male: Female, *n* (%)	20 (100): 0	30 (75): 10 (15)	15 (71): 6 (29)
Age (years), median (IQR)	46.5 (33–54)	49 (35–60)	38 (35–41)
TBSA (%), median (IQR)	6.5 (3–16)	9 (4–20)	
Days between burn and MRI acquisition, median (IQR)	81 (58–137)	70 (52–90)	
VAS, median (IQR)	7(6–8)	7 (6–8)	
BPI			
Sensory dimension, median (IQR)	23.9 (28–25)	22.4 (16–28)	
Reactive dimension, median (IQR)	41.9 (35–45)	38 (30–42)	
HRDS, median (IQR)	5 (2–6)	5 (3–6)	

NEB, non-electrical burn; EB, electrical burn; TBSA, total burn surface area; VAS, visual analog scale; BPI, Brief pain inventory; HRDS, Hamilton depression rating scale; values are presented as median (interquartile range, IQR).

**Table 2 jcm-10-02592-t002:** Clusters of increased or decreased CBV in all burn patients relative to controls.

Cluster	Voxel No.	Side	Brain Region	*t* Value	*p* Value	Peak Coordinates		
						*x*	*y*	*z*
Increased CBV in patients								
1	8	B	Postcentral gyrus	5.485	0.045	−4	−36	78
Decreased CBV in patients								
1	1044	B	Inferior frontal gyrus	5.408	0.004	−54	16	26
		B	Middle frontal gyrus	4.806	0.005	−54	16	32
		B	Precentral gyrus	4.227	0.008	−54	10	28
2	199	B	Superior temporal gyrus	4.766	0.031	−64	−16	−6
		B	Anterior lateral temporal lobe	4.673	0.026	−56	10	−8
		B	Inferior middle temporal gyrus	4.570	0.032	−66	−16	−6
3	59	B	Posterior temporal lobe	4.824	0.031	−40	−34	12
		B	Insula	4.650	0.035	−38	−34	12

CBV, cerebral blood volume; B, hemisphere associated with the burned limb

**Table 3 jcm-10-02592-t003:** Clusters of decreased CBV in EB group relative to controls.

Cluster	Voxel No.	Side	Brain Region	*t* Value	*p* Value	Peak Coordinates		
						*x*	*y*	*z*
Decreased CBV in patients with EB								
1	387	B	Anterior lateral temporal lobe	5.233	0.014	−58	8	−24
		B	Inferior middle temporal gyrus	4.393	0.021	−56	−4	−34
2	63	B	Inferior frontal gyrus	3.987	0.044	−54	16	26
		B	Middle frontal gyrus	3.961	0.045	−54	22	26
3	30	B	Superior temporal gyrus	5.133	0.038	−40	−26	8

CBV, cerebral blood volume; EB, electrical burn; B, hemisphere associated with the burned limb

**Table 4 jcm-10-02592-t004:** Clusters of increased or decreased CBV in NEB group relative to controls.

Cluster	Voxel No.	Side	Brain Region	*t* Value	*p* Value	Peak Coordinates		
						*x*	*y*	*z*
Increased CBV in patients with NEB								
1	132	B	Postcentral gyrus	5.838	0.017	−4	−38	78
		B	Superior parietal gyrus	5.477	0.018	−8	−56	72
Decreased CBV in patients with NEB								
1	213	B	Inferior frontal gyrus	5.262	0.017	−56	16	28
		B	Middle frontal gyrus	4.446	0.023	−54	16	32
		B	Precentral gyrus	4.081	0.031	−56	10	30
2	14	B	Middle frontal gyrus	4.324	0.046	−40	24	50

CBV, cerebral blood volume; NEB, non-electrical burn; B, hemisphere associated with the burned limb

## Data Availability

The data presented in this study are available on request from the corresponding author. The data are not publicly available due to the participants’ sensitive personal information.

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
