# Peer review of "Plastic Changes in Pain and Motor Network Induced by Chronic Burn Pain"

_jcm, 2021, doi:10.3390/jcm10122592_

Round 1

Reviewer 1 Report

Joo and colleagues examine neural changes associated with pain in individuals with electrical and non-electric burn compared to health controls. Using steady-state gadolinium enhanced MRI, cerebral blood volume analyses reveal altered brain function in brain areas commonly associated with pain and motion. The findings presented are of value, however its impact is not clear given the limited background on the topic and description of methods.

As mentioned, my main concern with the current manuscript is the limited information regarding the processing and statistical analysis of MRI data, which diminish my enthusiasm in the findings.

The authors state that pre- and post-contrast images were spatially transformed to the same standard space, yet the standardize space isn’t provided nor are the voxel dimensions after the transformation. 

The authors state in the last sentence of the introduction that “The purpose of the study was to analyze changes in CBV of the cerebral pain network”, yet it’s not clear in the methods if the analyses included voxels across the whole brain or only focused on the pain network.  If the goal is to look at pain network changes, why include voxels across the whole brain?  Alternatively, if the analysis focused on only pain/motor regions in the brain, then they need to be clearly defined.

The authors state that processing of CBV data was performed in SPM12 which included the randomize tool (Winkler et al., 2014). This is confusing because to my knowledge the randomize tool is part of the FSL library and SnPM toolbox, not the SPM.  Further, it is not clear how many permutations were run or which setting were used for the non-parametric permutation approach employed. All of this information is particularly crucial for evaluating the rate of type-1 errors in MRI studies that are on the smaller sample size such as this.

Minor comments:

  • A brief review of neuroimaging studies of pain or chronic pain would be helpful in the introduction.
  • The time from burn injury to study time is not provided or considered with regard to brain changes. Thus, the study claims they are investigating chronic pain without defining ‘chronic’.
  • One issue not addressed is how the current EB and NEB sample compares to general chronic pain samples. 
  • P values provided in the results and tables should be corrected p-values. It currently is not clear to the reader.
  • It is confusing to state that CBV was measured as an increase and decrease in the pain and motor network. Listing which regions exhibited increases and decreases would add more clarity.
  • Information on participant medication status would be helpful.
  • It’s not clear how this study’s findings compare with other brain imaging studies of pain.
  • There are a number of grammatically errors throughout the manuscript that make it hard to understand.

Author Response

  1. The authors state in the last sentence of the introduction that “The purpose of the study was to analyze changes in CBV of the cerebral pain network”, yet it’s not clear in the methods if the analyses included voxels across the whole brain or only focused on the pain network. If the goal is to look at pain network changes, why include voxels across the whole brain?  Alternatively, if the analysis focused on only pain/motor regions in the brain, then they need to be clearly defined.Answer> We agree with the reviewer. In this study, we analyzed all areas of brain anatomy, and analyzed meaningful areas in association with pain areas. Contents have been added to the introduction and discussion sections to clarify the reader’s understanding of this article.
  2. As mentioned, my main concern with the current manuscript is the limited information regarding the processing and statistical analysis of MRI data, which diminish my enthusiasm in the findings.The authors state that pre- and post-contrast images were spatially transformed to the same standard space, yet the standardize space isn’t provided nor are the voxel dimensions after the transformation. The authors state that processing of CBV data was performed in SPM12 which included the randomize tool (Winkler et al., 2014). This is confusing because to my knowledge the randomize tool is part of the FSL library and SnPM toolbox, not the SPM.  Further, it is not clear how many permutations were run or which setting were used for the non-parametric permutation approach employed. All of this information is particularly crucial for evaluating the rate of type-1 errors in MRI studies that are on the smaller sample size such as this.

    Answer> We appreciate you careful advise. Information on brain imaging analysis has been added to the method section. We hope that this description will help readers understand.

  3. The time from burn injury to study time is not provided or considered with regard to brain changes. Thus, the study claims they are investigating chronic pain without defining ‘chronic’. Answer> We agree with the reviewer. Unlike other musculoskeletal diseases, burn injuries can be distinguished from acute and chronic stages based on skin epithelialization during wound healing process. We specified the period in the inclusion criteria in the method section.
  4. One issue not addressed is how the current EB and NEB sample compares to general chronic pain samples. Answer> We agree with the reviewer. We added an explanation to the inclusion criteria in the method section to clearly explain the sample recruitment, and it was confirmed that there was no difference in the degree of deterioration due to burns and the intensity fo pain in the EB and NEB sample in the result section.
  5. P values provided in the results and tables should be corrected p-values. It currently is not clear to the reader.It is confusing to state that CBV was measured as an increase and decrease in the pain and motor network. Listing which regions exhibited increases and decreases would add more clarity. Answer> We agree with the reviewer. In the table 2,3,and 4, the increased or decreased brain areas are simply expressed clearly. We hope that the table arrangement will give readers a clearer understanding of the results of this article.
  6. Information on participant medication status would be helpful. Answer> We agree with the reviewer. As you pointed out, information on medication use has been added to the method section.
  7. It’s not clear how this study’s findings compare with other brain imaging studies of pain. Answer> We appreciate you careful advise. Pain caused by burn scar is the most common chronic complication and is a factor that interferes with daily life of burn patients. Until now, the mechanism of development related to the peripheral nervous system related to scars is well known, but the central nervous system has not been proven, so this is the first study to identify the central nervous system in patients with pain caused by burns. This study is expected to provide pain relief if a treatment method capable of recovering changes in the central nervous system is attempted in burn patients.
  8. There are a number of grammatically errors throughout the manuscript that make it hard to understand. Answer> We appreciate you careful advise. As you pointed out, the article was edited throughout the manuscript to native speakers. We highlight the corrected sentences.

Reviewer 2 Report

The authors have sought to identify and characterize any cerebral changes as a result of burn injury by means of cerebral blood volume measurements. The authors have done a very good job in clearly demonstrating cerebral changes after burn injury relative to baseline. I would, however, suggest the following revisions to the manuscript for clarity, specifically regarding the authors’ conclusion regarding the lack of differences between the two groups.

1) The authors conclude that there is not statistical significance between the EB and NEB groups. However, this is a slightly confusing statement the way the manuscript reads currently.

The authors state that in the EB group, CBV measurements were decreased in the pain and motor network, whereas in the NEB group, CBV was measured as "an increase and decrease in the pain and motor network". I interpret this to mean that the authors observed increases in some areas of the network and decreases in other areas of the network for patients in the NEB group. In addition, after stating this, the authors state in the next sentence that the CBV changes between the EB and NEB groups are not different.

This is confusing. It seems that, for example, there were patients in the NEB group that exhibited increases in CBV signal in the postcentral gyrus and superior parietal gyrus. These changes were given significant p values (relative to control, I assume). However, these changes were not seen in the EB group. In addition, the way the data is presented, the decrease in CBV seen in the EB group in the anterior lateral temporal lobe, inferior middle temporal gyrus and inferior frontal gyrus, was not observed in the NEB group. It would be useful for the authors to clarify how they came to the conclusion that there is not a difference when it seems like there are cortical changes unique to each group. If this is a statistical reason, perhaps a few sentences can be used to explain this analysis. Was this conclusion reached because the study did not have enough subjects, and thus statistical power to reach significance even though it trended in that direction? Or was this statement meant to highlight the fact that both EB and NEB only elicited changes in the sensory-discriminative dimension, and not in the emotional or cognitive dimension. If this is the case, a sentence should be added to clarify that in the summary at the end of the introduction.

2) It would be useful to have at least a brief mention early in the manuscript about the importance or reason for studying electrical burn patients specifically, and why the authors decided it is worthwhile to separate this subgroup of burns from all other types/sources of burn injury. The authors do not really elaborate on this until the discussion, but it would be useful to set-up this concept in the background section.

3) The authors mention in the background section that chronic pain is known to cause CNS changes, and have an interesting segment on involvement of the emotional dimension on chronic pain patients with depression. The authors do not discuss, however, how/if the results from their work are similar to, or different from, observed CNS changes in patients with other sources of chronic pain. This might be an interesting point to include.

4) Methods section (minor point): the authors state that the patients met both inclusion and exclusion criteria. Was this meant to say that the patients met the inclusion criteria, but none of the exclusion criteria? This is a minor detail in wording, but a slight alteration might improve clarity.

Author Response

  1. I would, however, suggest the following revisions to the manuscript for clarity, specifically regarding the authors’ conclusion regarding the lack of differences between the two groups. The authors conclude that there is not statistical significance between the EB and NEB groups. However, this is a slightly confusing statement the way the manuscript reads currently. 

    It would be useful for the authors to clarify how they came to the conclusion that there is not a difference when it seems like there are cortical changes unique to each group. If this is a statistical reason, perhaps a few sentences can be used to explain this analysis. Was this conclusion reached because the study did not have enough subjects, and thus statistical power to reach significance even though it trended in that direction? Or was this statement meant to highlight the fact that both EB and NEB only elicited changes in the sensory-discriminative dimension, and not in the emotional or cognitive dimension. If this is the case, a sentence should be added to clarify that in the summary at the end of the introduction.

    It would be useful to have at least a brief mention early in the manuscript about the importance or reason for studying electrical burn patients specifically, and why the authors decided it is worthwhile to separate this subgroup of burns from all other types/sources of burn injury. The authors do not really elaborate on this until the discussion, but it would be useful to set-up this concept in the background section. 

    Answer> We appreciate you careful advise. This article is a study to observe changes in the central nervous system in patients who maintain severe pain in burn scars after acute treatment. Groups were separated and observed because of the existence of other mechanism that may cause pain in electrical burns. According to the inclusion and exclusion criteria mentioned in the method section, as a purpose of observing the effect of only burn scars without severe dysfunction or depression, we could not observe the difference between electrical burn and non-electrical burn groups. 

    We mentioned that a comprehensive study including patients with severe dysfunction as well as patients with depression is needed to confirm changes in the central nervous system in all patients with electrical and non-electrical burns. We have added content to the discussion and limitation section on the hypotheses and the reasons for reaching different conclusions.

  2. Methods section (minor point): the authors state that the patients met both inclusion and exclusion criteria. Was this meant to say that the patients met the inclusion criteria, but none of the exclusion criteria? This is a minor detail in wording, but a slight alteration might improve clarity. Answer> We agree with the reviewer. We added the descriptions in the inclusion and exclusion criteria.

Round 2

Reviewer 1 Report

The authors have adequately addressed my concerns.